# A Novel Process to Recover Gypsum from Phosphogypsum

**DOI:** 10.3390/ma15051944

**Published:** 2022-03-05

**Authors:** Junhui Xiao, Tao Lu, Yuanfa Zhuang, Huang Jin

**Affiliations:** 1Sichuan Provincial Engineering Lab of Non-Metallic Mineral Powder Modification and High-Value Utilization, Southwest University of Science and Technology, Mianyang 621010, China; 2Postdoctoral Research Station, Dongfang Boiler Group Co., Ltd., Chengdu 611731, China; lut@dbc.com.cn (T.L.); zhuangyf@dbc.com (Y.Z.); jinh@dbc.com (H.J.); 3Key Laboratory of Ministry of Education for Solid Waste Treatment and Resource Recycle, Southwest University of Science and Technology, Mianyang 621010, China; 4Institute of Multipurpose Utilization of Mineral Resources, Chinese Academy of Geological Sciences, Chengdu 610041, China

**Keywords:** phosphogypsum, gypsum, classification, flotation

## Abstract

In this study, we investigated a coarse phosphogypsum containing 49.63% SO_3_, 41.41% CaO, 10.68%, 4.47% SiO_2_, 1.28% P_2_O_5_, 0.11% F, CaSO_4_·2H_2_O purity of 80.65%, and whiteness of 27.68. Phosphogypsum contains calcium sulfate dehydrate as the main mineral, with small amounts of brushite, quartz, muscovite, and zoisite. Harmful elements, such as silicon, phosphorus, and fluorine, are mainly concentrated in the +0.15 mm and −0.025 mm fraction, which can be pre-selected and removed by the grading method to further increase the CaSO_4_·2H_2_O content. Gypsum was recovered using a direct flotation method, which included one roughing, one scavenging, and two cleaning operations, from −0.15 mm to +0.025 mm. The test results show that a gypsum concentrate with a CaSO_4_·2H_2_O purity of 98.94%, CaSO_4_·2H_2_O recovery of 80.02%, and whiteness of 37.05 was achieved. The main mineral in the gypsum concentrate was gypsum, and limited amounts of muscovite and zoisite entered the gypsum concentrate because of the mechanical entrainment of the flotation process.

## 1. Introduction

Phosphogypsum is mainly obtained from the phosphate fertilizer industry and is a solid waste residue produced by a wet preparation of phosphoric acid. The global cumulative emissions of phosphogypsum total approximately 6 billion tons, which is increasing at a rate of 150 million tons/year. It is projected that the total amount of phosphogypsum will double by 2025–2045 [1,2]. Phosphogypsum is an industrial by-product of gypsum with a low utilization rate owing to its large discharge, complex impurity composition, and difficult treatment. The accumulation of large amounts of phosphogypsum can cause environmental risks. Therefore, effective treatment and efficient utilization of phosphogypsum are urgently needed. The main impurities in phosphogypsum are divided into three categories: phosphorus, fluorine, and organic impurities. Phosphorus impurities in phosphogypsum mainly consist of soluble phosphorus (H_3_PO_4_, H_2_PO^4−^, and HPO_4_^2−^), insoluble phosphorus (Ca_3_(PO_4_)_2_), and eutectic phosphorus (CaHPO_4_·2H_2_O). The fluorine-containing impurities in phosphogypsum mainly consist of soluble fluorine (NaF and KF) and insoluble fluorine (Na_3_AlF_6_, CaSiF_6_, and CaF_2_). Organic impurities in phosphogypsum mainly consist of organic matter inherent in phosphate rock and organic additives added in the production process [3,4,5]. The comprehensive utilization rate of phosphogypsum is low, owing to the various complex impurities and the difficult occurrence state. Presently, the main separation and purification methods for phosphogypsum are as follows.

Phosphorus and fluorine in phosphogypsum are largely soluble in water. Owing to their solubility in water, they can be effectively removed by rinsing, filtering, leaching, and dehydration. However, the single water-washing process consumes a large amount of water and a high amount of energy. The newly generated wastewater causes secondary pollution, and the wastewater needs to be treated separately to meet the national discharge standards before discharge, which significantly increases the treatment cost. Therefore, this process has not been applied on a large scale. Moreover, it is necessary to realize stepwise water recycling through technological improvements, reduce water consumption, and recycle wastewater-soluble impurities, such as phosphorus and fluorine, in green and low-cost ways [6,7,8].

Organic impurities can be removed by conventional flotation. This process includes pouring phosphogypsum and water into flotation equipment in the right proportions and using the natural floating of organic impurities to scrape off the impurities. This process is suitable for treating phosphogypsum with a high organic content and can improve the whiteness of phosphogypsum; however, this method has low efficiency and no significant removal effect on soluble impurities. Due to the fact that the water used in the flotation process can be recycled, it is often combined with the water-washing process. Presently, the purification of phosphogypsum by adding a flotation agent has also been widely studied. This treatment method involves the addition of alkaline-modified materials, such as quick lime, into phosphogypsum; the alkaline-modified material reacts with the soluble phosphorus and soluble fluorine and converts the refractory inert materials to precipitates. This method can homogenize phosphogypsum with high-quality fluctuations and low organic matter content. Lime neutralization treatment is widely used in the production of cement retarders because of its simple process, low investment, obvious effect, and low amount of secondary pollution. However, this method can only temporarily solve the harmful effects of soluble phosphorus and fluorine. Soluble phosphorus and fluorine precipitate after a long time, and this method cannot remove the adverse effects of organic matter on phosphogypsum. In the process of calcination at 800 °C, the P_2_O_5_ in phosphogypsum is converted into stable inert phosphate, and a small amount of organophosphorus and hydrogen fluoride can be removed by volatilization. Unlike other calcination processes, flash sintering aims to transform soluble phosphorus and fluorine into inert substances without water washing. The flash-burning method is usually combined with the lime neutralization method to avoid the volatilization of fluoride, which pollutes the environment and causes secondary pollution; however, this method has high energy consumption, large initial investment, and its equipment is prone to corrosion [9,10,11,12,13].

In summary, extensive research on phosphogypsum pretreatment has been performed by scientists and technicians. However, each method has some disadvantages, and there are still some gaps in large-scale industrial applications. Thus, it is necessary to combine various pretreatment methods to learn from each of them, which is an important research direction in the future. In addition, the authors believe that flotation is a promising method. Organic impurities surfaced by reverse flotation, CaSO_4_·2H_2_O surfaced by forward flotation, and then a small amount of lime phosphorus fixation and fluorine were added to the forward-flotation-filtered concentrate, such that phosphogypsum could be completely purified at a low cost. However, further experiments are required to verify this hypothesis.

## 2. Materials and Methods

### 2.1. Materials

The test samples used in this study were taken from phosphogypsum produced by a phosphorus chemical enterprise in the Deyang Region, Sichuan Province, China. The content of CaSO_4_·2H_2_O in the phosphogypsum was 80.65%, and the whiteness was 27.68. The water content of the phosphogypsum was less than 5%, and the particle size was less than 1 mm. The main chemical compositions of the samples are shown in Table 1.

The main chemical reagents used in this test were sulfuric acid, sodium silicate, sodium hexametaphosphate, starch, carboxymethylcellulose, dodecylamine, mixed amine, octadecylamine, dodecyltrimethylammonium chloride, and pine oil. All reagents were of analytical grade and were obtained from a producing area in Tianjin Tianli Chemical Reagent Co., Ltd., Tianjin, China.

### 2.2. Experiment

Flotation (roughing, scavenging) aimed at improving the recovery of gypsum concentrate was performed using an XFD-1.5 L hanging tank flotation machine (Jinlin Exploration Machinery Plant, Changchun, China) operating at a spindle speed of 1650 r/min. A 500 g mass of phosphogypsum was added to the 1.5 L flotation tank. Flotation (cleaning) aimed at further increasing the purity and whiteness of the gypsum concentrate was performed using an XFD-1.0 L hanging tank flotation machine (Jinlin Exploration Machinery Plant, Changchun, China) operating at a spindle speed of 1,650 r/min. A 300 g mass of gypsum concentrate was added to the 1.0 L flotation tank. Distilled water (1.0 L) was added, and the pulp was stirred and mixed for 3 min, followed by adjustment to the required pH using sulfuric acid. After 5 min of pulping, the depressant was added to the slurry and conditioned for 3 min. The collectors were then added and agitated for 3 min. Before aeration, frothers (pine oil) were added to improve the bubbles and stirred for an additional 3 min. After 3 min of flotation, the froth (gypsum concentrate) and in-tank product (flotation tailings) were separately filtered, the samples were filtered, dried at 40 °C, weighed, and the gypsum recovery was calculated according to Equation (1).
*R* = (Q_1_ × G_1_)/(Q_0_ × G_0_),(1)
where *R* is the recovery of gypsum (%), Q_1_ is the weight of the flotation concentrate (g), G_1_ is the calcium sulfate dehydrate of the flotation concentrate, Q_0_ is the weight of phosphogypsum (g), and G_0_ is the calcium sulfate dehydrate of phosphogypsum (g).

### 2.3. Analyses

The whiteness of the gypsum concentrate samples was measured using a WSD-3C whiteness instrument manufactured by Beijing Kangguang Optical Instrument Co., Ltd. The equipment was preheated for 30 min before the test, and calibration was subsequently performed using a standard colorimetric plate. After calibration, the samples were tested. The purity of the calcium sulfate dihydrate in the phosphogypsum was determined by the content of crystallized water in the gypsum. The mass fraction of calcium sulfate dihydrate (*G*) in the sample was calculated using Equation (1). Gypsum recovery was calculated using Equation (2):*G* = 4.7885 × *H*,(2)
where *G* is the purity of calcium sulfate dihydrate (%), and 4.7785 is the coefficient of calcium sulfate dehydrate content converted from crystal water content, which is equal to the molecular weight of the calcium sulfate dihydrate divided by the molecular weight of two water molecules. *H* represents the crystal water content (%).

The chemical compositions of the solid materials (including phosphogypsum, gypsum concentrate, and flotation tailings) were analyzed using a Z–2000 atomic absorption spectrophotometer (Hitachi Co., Ltd., Tokyo, Japan). The mineral phase compositions of the aforementioned solid substances were examined using an XRD (X’ Pert Pro, Panaco, The Netherlands). The microstructures of the solid products were investigated using scanning electron microscopy (SEM, S440, Hirschmann Laborgerate GmbH & Co. KG, Eberstadt, Germany) equipped with an energy-dispersive X-ray spectroscopy (EDS) detector (UItra55, CarlzeissNTS GmbH, Hirschmann Laborgerate GmbH & Co. KG, Eberstadt, Germany).

## 3. Results

### 3.1. Process Mineralogical Analysis of Phosphogypsum

Figure 1a shows the X-ray diffraction (XRD) patterns of the samples. Phosphogypsum mainly contains calcium sulfate dehydrate (CaSO_4_·2H_2_O), along with small amounts of CaPO_3_(OH)·2H_2_O, SiO_2_, Kal_2_SiO_10_(OH), and Ca_2_Al_3_[SiO_12_]OH. This indicates that the crystalline phase of phosphogypsum is mainly gypsum, and the main mineral impurities are brushite and silicaluminate minerals, which need to be removed in the subsequent purification. The results in Table 2 and Figure 2b show that, compared with the content of SiO_2_, F, and P_2_O_5_ in different fractions, those in the +0.15 mm fraction were 12.89%, 0.41%, and 5.08%, respectively, whereas those in the −0.025 mm fraction were 13.17%, 0.35%, and 1.56%, respectively. This further indicates that the impurity elements, i.e., silicon, fluorine, and phosphorus, are mainly concentrated in the +0.15 mm and −0.025 mm fractions. Therefore, in this study, the +0.15 mm and −0.025 mm products with two particle sizes were pre-selected by a classification method, and gypsum was further recovered from the −0.15 and +0.025 mm raw materials by flotation.

### 3.2. Effects of Flotation Flowsheet Parameters on Gypsum Purification

#### 3.2.1. Collectors Dosages

A study on the surface properties of gypsum showed that the gypsum surface was charged, and the cationic collector was easily adsorbed onto its surface. Amine flotation agents such as dodecylamine and quaternary ammonium salt are common cationic collectors that are widely used in numerous mineral separations [14,15,16,17]. The effects of dodecylamine, octadecylamine, mixed amine, and dodecyltrimethylammonium chloride on the separation and purification of gypsum were investigated under the following test conditions: flotation concentration of 25%, flotation pulp pH = 2.5 using sulfuric acid, a pine dosage of 40 g/t; the results are shown in Figure 2a–c). Using the same amount of collector, the order of collecting abilities of gypsum was found to be: mixed amine > dodecylamine > dodecyltrimethylammonium chloride > octadecylamine. With an increase in the dosage of MA, the purity of CaSO_4_·2H_2_O in the concentrate decreased gradually, and the recovery of gypsum increased gradually. When the dosage of mixed amine reached 200 g/t and continued to increase, the purity of CaSO_4_·2H_2_O and whiteness of the concentrate decreased significantly, mainly owing to the increase in the dosage of the reagents, resulting in an increased floatation of the concentrate and the easy inclusion of other impurities. Regarding the cost of the mixing of reagents, a mixed amine dosage of 200 g/t is suitable for flotation. The purity of CaSO_4_·2H_2_O in the concentrate was 95.89%, the recovery of CaSO_4_·2H_2_O was 79.93%, and the whiteness was 35.63. 

#### 3.2.2. Depressants Dosages

Sodium silicate, sodium hexametaphosphate, starch, and carboxymethylcellulose were used as flotation agents for gangue minerals [18,19,20], and the effects of different dosages on gypsum flotation were investigated under the following test conditions: flotation concentration of 25%, flotation pulp pH = 2.5 using sulfuric acid, mixed amine dosage of 200 g/t, and pine dosage of 40 g/t. The results in Figure 3a–c show that sodium silicate, sodium hexametaphosphate, and carboxymethylcellulose influence gangue minerals. However, the addition of ST can reduce the purity of CaSO_4_·2H_2_O, the recovery rate of CaSO_4_·2H_2_O, and the whiteness of the concentrate, which further indicates that the inhibitory effect of the four inhibitors on gangue minerals is as follows: sodium silicate > carboxymethylcellulose > sodium hexametaphosphate > starch. Sodium silicate can be selectively adsorbed onto the surface of gangue minerals, this results in an increase in the hydrophilicity of gangue minerals and changes to the surface electrical properties of gangue minerals. Thus, the gangue minerals and gypsum surfaces are negative, increasing the electrostatic repulsion between gypsum and gangue minerals, reducing the agglomeration of mineral particles, and improving the dispersion of the flotation system [21,22,23]. An increase in the dosage of sodium silicate improved the quality of the concentrate; however, when the dosage of sodium silicate exceeded 400 g/t, the purity of CaSO_4_·2H_2_O and the whiteness of the concentrate decreased, and the recovery of CaSO_4_·2H_2_O increased to a certain extent. Therefore, a sodium silicate dosage of 400 g/t is ideal, and the purity of CaSO_4_·2H_2_O in the concentrate increased to 96.38%, the whiteness increased to 36.63, and the recovery of CaSO_4_·2H_2_O increased to 81.51%.

#### 3.2.3. pH Value of Pulp

Different flotation pH values of the pulp tests were conducted under the following conditions: flotation concentration of 25%, mixed amine dosage of 200 g/t, sodium silicate dosage of 400 g/t, and pine dosage of 40 g/t; the results are shown in Figure 4a–c. When the pH of the pulp is low (pH = 1.5–2.0), the separation of gypsum and quartz is different, and the separation effect of gypsum and quartz is better. When the pH of the pulp is higher (pH > 2.0), the difference in separation between gypsum and quartz becomes small, and the separation effect of gypsum and quartz is poor. This is because the isoelectric point of gypsum is pH = 1–2, and that of quartz is pH = 2.3–3.0. When the pH of the slurry was higher than 2.3, both the gypsum and the quartz minerals were negatively charged on the surface, which could be adsorbed by the cationic collector and showed good floatability; thus, they could not be separated. When the pulp pH is ≤2, the gypsum mineral surface with a negative charge can exhibit cationic collector electrostatic adsorption, whereas a quartz surface with positive charge or no charge cannot exhibit cationic collector adsorption, to achieve gypsum and quartz separation. A gypsum surface potential of PZC = 2.3 is negatively charged over a wide pH range, and a cationic collector can easily be adsorbed onto the gypsum surface. The molecular layer of the gypsum crystal water and the Ca–O bond are weak and easy to dissociate from these positions when broken, which also leads to natural hydrophilic gypsum. It is also easy to achieve the separation of hydrophobic organic matter and other impurities from phosphogypsum and gypsum by adding a foaming agent [24,25,26,27,28]. After comprehensive consideration, the purity, whiteness, and recovery of CaSO_4_·2H_2_O in the concentrate were 96.38%, 36.63, and 84.08%, respectively, at pH = 2.0.

#### 3.2.4. Flotation Concentration

The effects of different dosages on gypsum flotation were investigated under the following conditions: pH = 2.0 (H_2_SO_4_), mixed amine dosage of 200 g/t, sodium silicate dosage of 400 g/t, and pine dosage of 40 g/t; the results are shown in Figure 5a–c. Flotation concentration is an important process parameter that influences the beneficiation index, which mainly influences the pulp filling capacity, the concentration of reagents in the pulp, and the flotation time. Generally, a proper increase in the concentration of the floating material can reduce the probability of particles (especially coarse particles) falling off the bubble, thus improving the recovery of useful minerals from coarse particles. A high concentration of the material plays an important role in weakening the influence of density and particle size on the velocity of the upper floating end [29,30,31,32,33,34]. However, owing to the fine particle size of the phosphogypsum material, the CaSO_4_·2H_2_O content in the concentrate decreased significantly; the whiteness of the concentrate also decreased when the concentration exceeded 30%, and the recovery of the CaSO_4_·2H_2_O increased slightly. Therefore, 30% is the ideal selection of flotation concentration, and the purity and recovery of CaSO_4_·2H_2_O in the concentrate are 96.18% and 84.16%, respectively, and the whiteness of the concentrate is 36.28.

#### 3.2.5. Time of Flotation Scavenging

To further improve the comprehensive recovery rate of gypsum from phosphogypsum, scavenging tests were performed to determine suitable times for the recovery of gypsum. The test conditions for the roughing flotation were a flotation concentration of 30%, pH = 2.0 (H_2_SO_4_), mixed amine dosage of 400 g/t, sodium silicate dosage of 400 g/t, and pine dosage of 20 g/t. The test conditions for scavenging I were MA dosage of 200 g/t, pine dosage of 20 g/t, and the test conditions of scavenging II were MA dosage of 100 g/t and pine dosage of 10 g/t. The technological process is illustrated in Figure 6, and the results are listed in Table 3. 

An increase in the time of scavenging can improve the recovery rate of gypsum; however, after two scavenging operations, the whiteness of the gypsum concentrate decreased significantly, and, after two cleaning operations, the whiteness reduced to 29.63%. Owing to the increase in scavenging times, the consumption of flotation reagents increased correspondingly. In the case of the flotation process with one roughing and one scavenging step, a gypsum concentrate with a CaSO_4_·2H_2_O purity of 96.43%, a whiteness of 35.94, and a CaSO_4_·2H_2_O recovery of 83.72% can be obtained. 

#### 3.2.6. Time of Flotation Cleaning

Gypsum concentrate with a CaSO_4_·2H_2_O purity of 96.43%, whiteness of 35.94, and CaSO_4_·2H_2_O recovery of 83.72% was obtained from the effects of flotation scavenging, using the flotation process of one roughing and one scavenging step. The technological process of flotation cleaning shown in Figure 7 was used to further purify the gypsum concentrate; the test conditions of the roughing flotation were a flotation concentration of 30%, pH = 2.0 (H_2_SO_4_), a mixed amine dosage 400 g/t, a sodium silicate dosage of 400 g/t, and a pine dosage of 40 g/t; the test conditions of scavenging were an MA of dosage 200 g/t and pine dosage of 20 g/t. The results in Table 4 indicate that a two-stage concentration process can improve the purity, whiteness, and recovery of the gypsum concentrate. The purity of the CaSO_4_·2H_2_O, the recovery of the CaSO_4_·2H_2_O, and the whiteness were 98.94%, 79.87%, and 37.05, respectively.

#### 3.2.7. The Entire Flowsheet Test of Recovering Gypsum from Phosphogypsum

The technological process of recovering gypsum from phosphogypsum was obtained using a single flotation process test, a scavenging test, and a cleaning test. To obtain the product index of the entire process, the process shown in Figure 8 was adopted to perform the entire flowsheet test. The results in Table 5 and Table 6 show that gypsum concentrates with a CaSO_4_·2H_2_O purity of 98.94%, a CaSO_4_·2H_2_O recovery of 80.02%, and a whiteness of 37.05 were obtained. The content of silicon, phosphorus, fluorine, and other impurities in the gypsum concentrate was relatively low; therefore, the obtained gypsum concentrate could be used as a high-quality raw material for the preparation of α-hemihydrate high-strength gypsum or β-hemihydrate building gypsum.

## 4. Discussion

Figure 9 and Figure 10 show that gypsum minerals and gangue minerals in untreated phosphogypsum adhere to each other, and fine mineral particles are attached to the surface of the gypsum crystals. The main mineral in the gypsum concentrate was gypsum (CaSO_4_·2H_2_O); however, the brushite (CaPO_3_(OH)·2H_2_O) mineral phase and quartz (SiO_2_) were not found. Meanwhile, tiny amounts of muscovite (KAl_2_SiO_10_(OH)) and zoisite (Ca_2_Al_3_[SiO_12_]OH) entered the gypsum concentrate owing to mechanical entrainment of the flotation process. Figure 11a–d shows further that the phosphogypsum, after grading and flotation of the gypsum concentrate, dispersed and had a smooth crystal surface, which indicates that the phosphogypsum through classification of pre-treatment using direct flotation can effectively remove impurities in the phosphogypsum concentrate, thus significantly improving the product quality of the gypsum concentrate.

## 5. Conclusions

The following are the main conclusions based on the results obtained in this study on the recovery of gypsum from phosphogypsum:(1)Phosphogypsum contained 80.65% CaSO_4_·2H_2_O and whiteness of 27.68. The main mineral is CaSO_4_·2H_2_O with small amounts of brushite, quartz, muscovite, and zoisite in phosphogypsum. Harmful elements, such as silicon, phosphorus, and fluorine, are mainly concentrated in the +0.15 mm and −0.025 mm fraction; these can be pre-selected and removed by the grading method to increase the CaSO_4_·2H_2_O content and reduce the processing cost.(2)A novel direct flotation process, consisting of one roughing, one scavenging, and two cleaning operations, were employed to recover gypsum from the −0.15 to + 0.025 mm fraction materials. The addition of NaSiO_3_ enhanced the removal of gangue minerals. Furthermore, the addition of a mixed amine enhanced the capture of CaSO_4_·2H_2_O. The test results show that gypsum with a CaSO_4_·2H_2_O content of 98.94%, CaSO_4_·2H_2_O recovery of 80.02%, and whiteness of 37.05 was obtained under the following conditions: flotation roughing concentration of 30%, pH = 2.0 (H_2_SO_4_), mixed amine dosage of 400 g/t, pine dosage of 40 g/t, flotation scavenging of mixed amine dosage of 200 g/t, sodium silicate dosage of 400 g/t, and pine dosage of 20 g/t. Gypsum concentrate could be used as a high-quality raw material to prepare α-hemihydrate high-strength gypsum or β-hemihydrate-building gypsum.(3)The results of XRD and SEM-EDS demonstrated that the main mineral in the gypsum concentrate was gypsum; however, brushite mineral phase and quartz were not found. Meanwhile, a tiny amount of muscovite and zoisite entered the gypsum concentrate owing to the mechanical entrainment of the flotation process.

## Figures and Tables

**Figure 1 materials-15-01944-f001:**
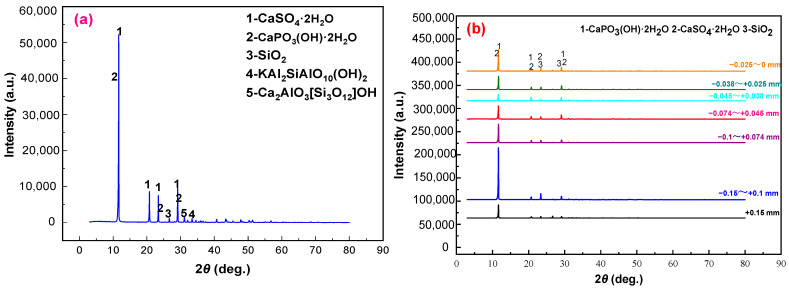
XRD patterns of the phosphogypsum (**a**) and the different grain-sized products (**b**).

**Figure 2 materials-15-01944-f002:**
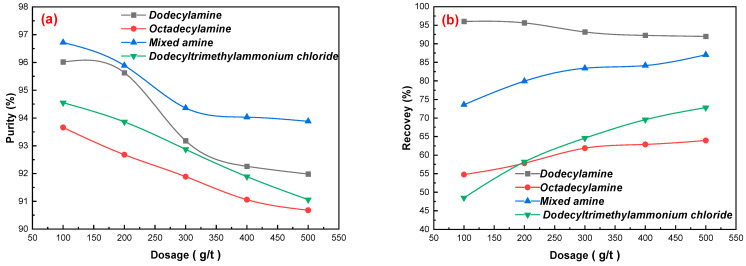
Effects of dosages of dodecylamine, octadecylamine, mixed amine, and dodecyltrimethylammonium chloride on gypsum purification: (**a**) gypsum purity, (**b**) gypsum recovery, and (**c**) gypsum whiteness.

**Figure 3 materials-15-01944-f003:**
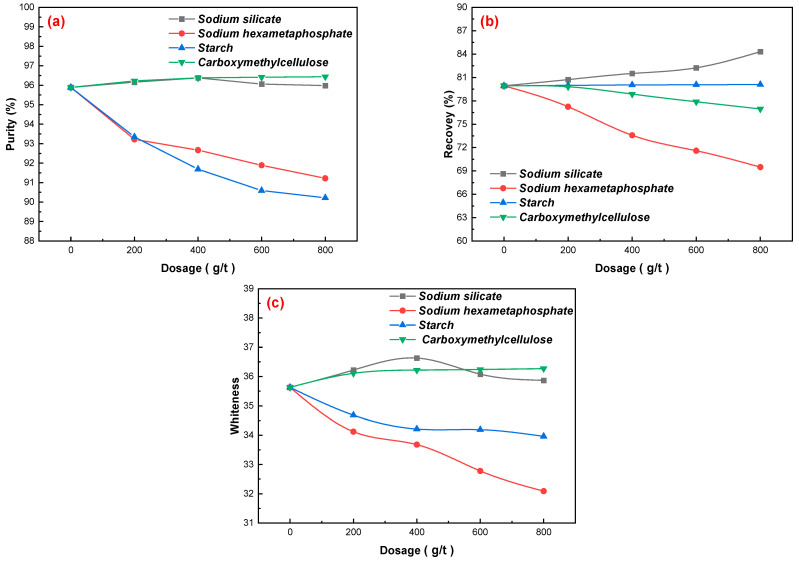
Effects of the dosages of sodium silicate, sodium hexametaphosphate, starch, and carboxymethylcellulose on gypsum purification: (**a**) gypsum purity, (**b**) gypsum recovery, and (**c**) gypsum whiteness.

**Figure 4 materials-15-01944-f004:**
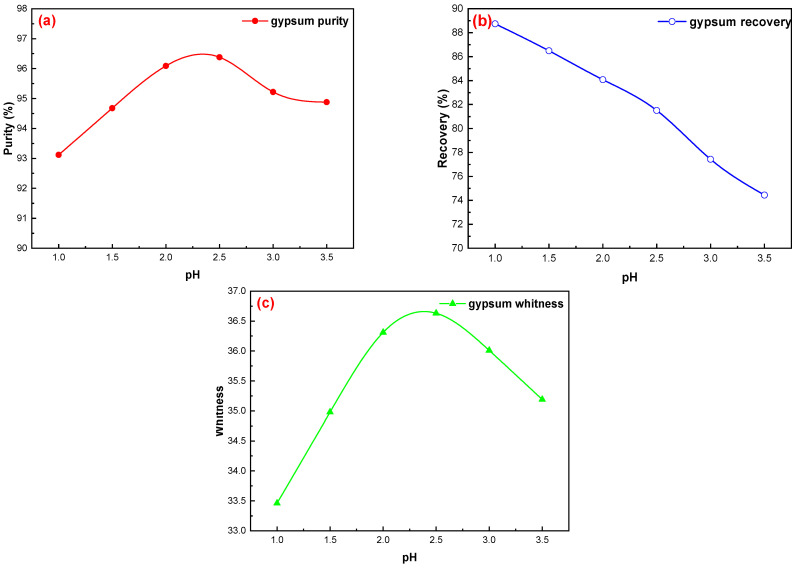
Effects of flotation pH value on gypsum purification: (**a**) gypsum purity, (**b**) gypsum recovery, and (**c**) gypsum whiteness.

**Figure 5 materials-15-01944-f005:**
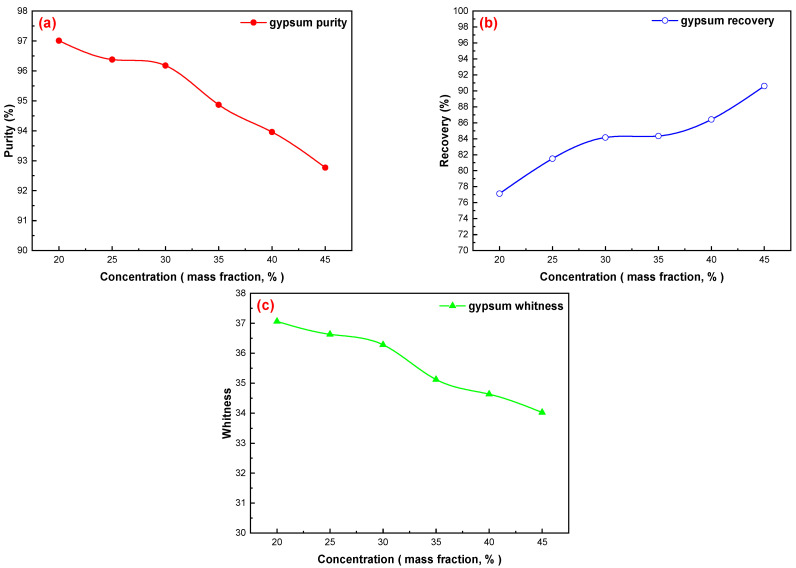
Effects of flotation concentration on gypsum purification: (**a**) gypsum purity, (**b**) gypsum recovery, and (**c**) gypsum whiteness.

**Figure 6 materials-15-01944-f006:**
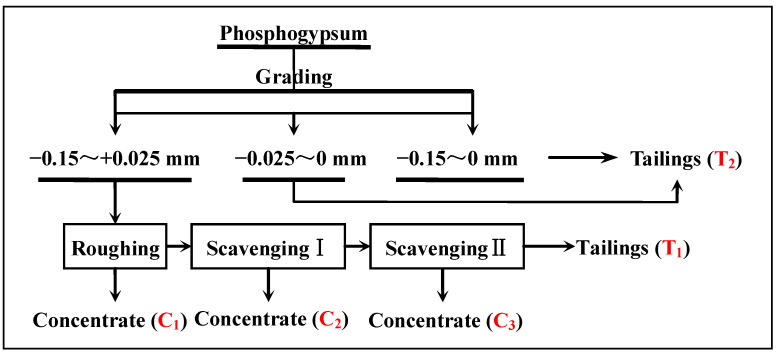
Scavenging test flowsheet of the recovery of gypsum from phosphogypsum.

**Figure 7 materials-15-01944-f007:**
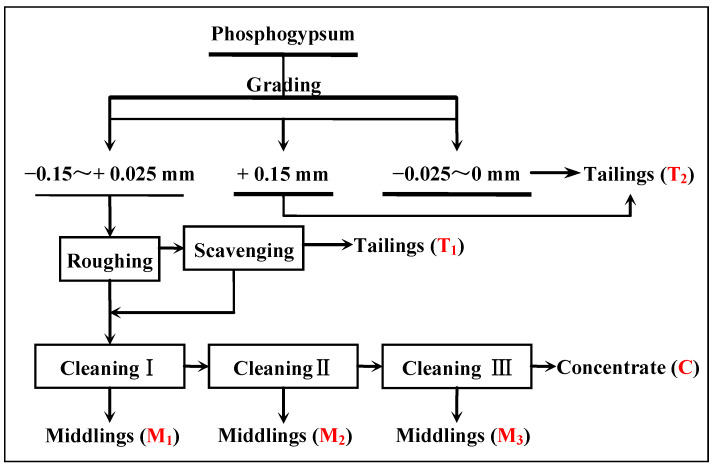
Cleaning test flowsheet of the recovery of gypsum from phosphogypsum.

**Figure 8 materials-15-01944-f008:**
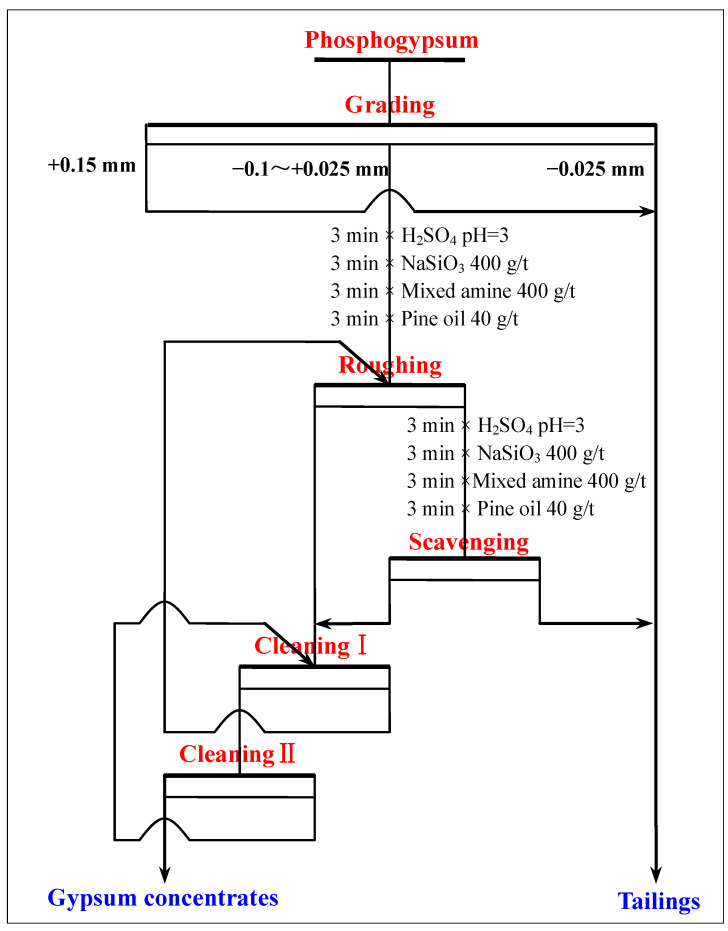
The recovery of gypsum from phosphogypsum using grading and flotation.

**Figure 9 materials-15-01944-f009:**
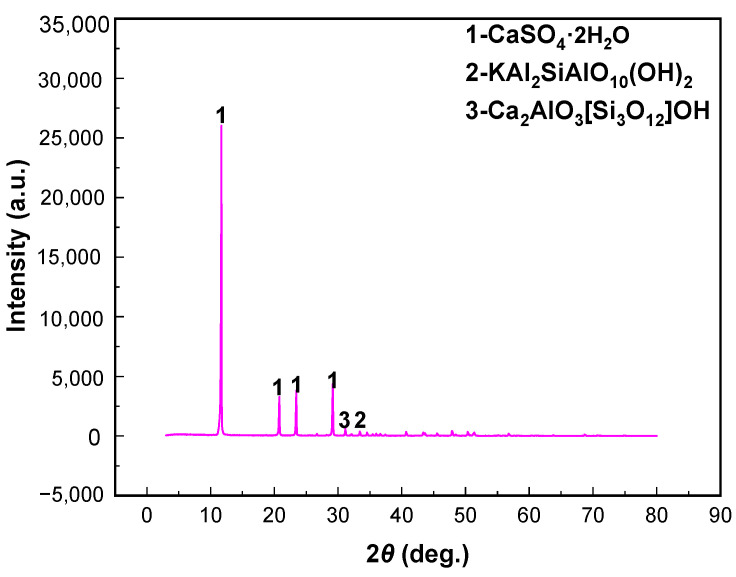
XRD patterns of gypsum concentrates.

**Figure 10 materials-15-01944-f010:**
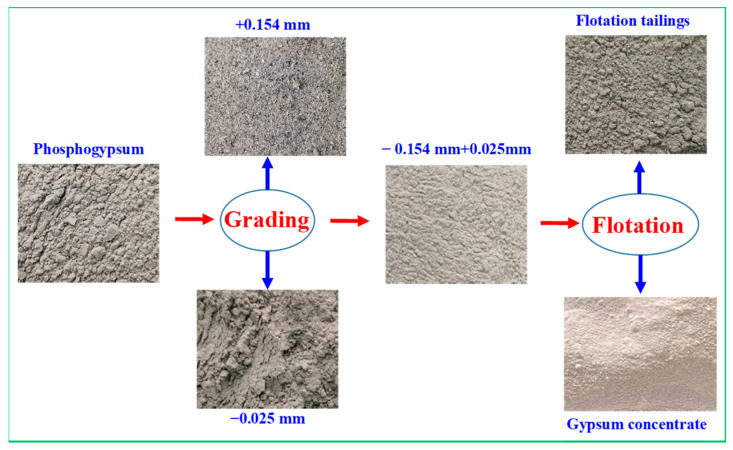
Photographs of characterization analysis of purification process of phosphogypsum.

**Figure 11 materials-15-01944-f011:**
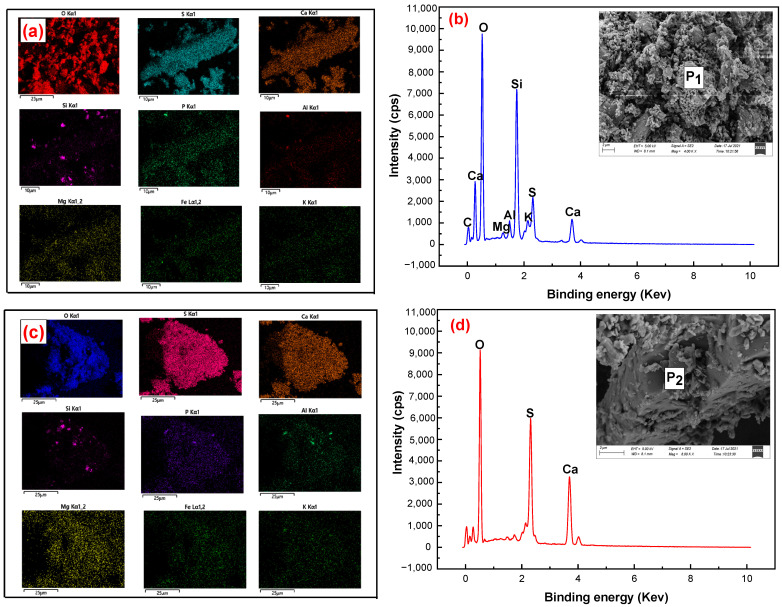
SEM-EDS analysis of (**a**,**b**) phosphogypsum, and (**c**,**d**) gypsum concentrates.

**Table 1 materials-15-01944-t001:** Main chemical composition analysis of phosphogypsum (mass fraction, %).

SO_3_	CaO	SiO_2_	P_2_O_5_	Al_2_O_3_	Fe_2_O_3_	K_2_O	SrO	MgO	F	TiO_2_	BaO	Na_2_O	Y_2_O_3_
49.62	41.41	5.47	1.28	0.63	0.31	0.20	0.12	0.12	0.11	0.10	0.07	0.05	0.01

**Table 2 materials-15-01944-t002:** Grain-sized analysis of phosphogypsum (mass fraction, %).

Fraction (mm)	Yield	SO_3_	CaO	SiO_2_	P_2_O_5_	Al_2_O_3_	Fe_2_O_3_	K_2_O	MgO	F
+0.15	7.60	41.17	38.35	12.89	5.08	1.00	0.57	0.34	0.10	0.41
−0.15~+0.1	11.21	51.89	43.23	2.93	1.15	0.28	0.21	0.08	0.02	0.004
−0.1~0.074	19.01	51.76	44.24	2.25	1.01	0.25	0.18	0.06	0.06	0.005
−0.074~0.045	14.23	52.08	44.09	2.15	1.03	0.19	0.26	0.05	0.02	0.006
−0.045~+0.038	17.35	51.86	43.93	2.34	1.03	0.26	0.28	0.05	0.02	0.005
−0.038~+0.025	10.26	51.09	43.39	3.24	1.16	0.40	0.35	0.08	0.03	0.003
−0.025~0	20.34	44.81	37.28	13.17	1.56	1.36	0.87	0.39	0.12	0.35
Totals	100.00	49.55	42.10	5.46	1.47	0.54	0.39	0.15	0.06	0.11

**Table 3 materials-15-01944-t003:** Scavenging test results of the recovery of gypsum from phosphogypsum.

Products	Mass Fraction (%)						Whiteness	
	Yield		Purity		Recovery		Individual	Cumulative
	Individual	Cumulative	Individual	Cumulative	Individual	Cumulative		
C_1_	62.54	62.54	96.45	96.45	74.72	74.72	36.28	36.28
C_2_	7.55	70.09	96.25	96.43	9.00	83.72	33.16	35.94
C_3_	0.76	70.85	92.16	96.38	0.87	84.59	29.63	35.88
T_1_	1.18	72.03	76.34	96.05	1.12	85.71	7.35	35.41
T_2_	27.97	100	41.23	80.72	14.29	100.00	8.15	27.78
Totals	100.00		80.72		100.00		27.78	

**Table 4 materials-15-01944-t004:** Cleaning test results of recovering gypsum from phosphogypsum.

Products	Mass Fraction (%)						Whiteness	
	Yield		Purity		Recovery		Individual	Cumulative
	Individual	Cumulative	Individual	Cumulative	Individual	Cumulative		
C	64.56	64.56	98.98	98.98	79.11	79.11	37.07	37.07
M_3_	0.19	64.75	95.88	98.97	0.22	79.33	35.75	37.07
M_2_	0.46	65.21	94.65	98.94	0.54	79.87	35.26	37.05
M_1_	0.96	66.17	93.12	98.86	1.11	80.98	32.37	36.98
T_1_	5.87	72.04	65.25	91.31	4.74	85.72	6.66	34.52
T_2_	27.96	100.00	41.24	80.77	14.28	100.00	8.67	27.72
Totals	100.00		80.77		100.00		27.72	

**Table 5 materials-15-01944-t005:** Entire flowsheet test results of the recovery of gypsum from phosphogypsum.

Products	Mass Fraction (%)			Whiteness
	Yield	Purity	Recovery	
Gypsum concentrates	65.27	98.92	80.02	37.12
Tailings	34.73	46.43	19.98	9.96
Totals	100.00	80.69	100.00	27.69

**Table 6 materials-15-01944-t006:** Main chemical composition analysis of gypsum concentrates (mass fraction, %).

SO_3_	CaO	SiO_2_	P_2_O_5_	Al_2_O_3_	Fe_2_O_3_	K_2_O	SrO	MgO	F	TiO_2_	BaO	Na_2_O	Y_2_O_3_
52.52	44.88	0.32	0.05	0.05	0.02	0.05	0.01	0.06	0.01	0.02	0.01	0.01	0.01

## Data Availability

Not applicable.

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
