# Peer review of "A Novel Process to Recover Gypsum from Phosphogypsum"

_materials, 2022, doi:10.3390/ma15051944_

Round 1

Reviewer 1 Report

Dera Authors,

All my comments are in the attached file.

I suggest changes to the content of the abstract. The abstract gives too much details of the chemical composition. It's unnecessarily. Additionally, these values are incorrect. The same applies to the details of the flotation conditions. The abstract should generally introduce the subject of the article.

Author Response

Respond to Reviewer 1# comments

Point 1: I suggest changes to the content of the abstract. The abstract gives too much details of the chemical composition. It's unnecessarily. Additionally, these values are incorrect. The same applies to the details of the flotation conditions. The abstract should generally introduce the subject of the article.

Respond to 1: Thank you for your careful review of the article. We have revised the abstract and simplified part of the content. At the same time, we have also made a lot of modifications in the document you marked, for which we are deeply grateful, which is very helpful to improve the quality of our articles, and we look forward to the next opportunity for learning and communication.

Thank you again for your valuable suggestions, which will be very helpful for our follow-up research. Any questions, please feel free to contact us, I would be very grateful.

With kind regards,

Dr. Junhui Xiao

Reviewer 2 Report

A Novel Process to Recover Gypsum from Phosphogypsume is very interesting paper. Some improvement is required.

Line 2: Please to correct "Phosphogypsume" to Phosphorgypsum.

Line 3:  Huang Jin 2 Firstname Lastname 1  (delete Firstname, LAstname)

Line 13: whiteness of 27.68. (what is the content of rare earth elements and radioactive elements such as thorium and uranium)

Line 169: Agents such as DDA (please to write full name)

Line 171: MA, and DTAC (please to write full name)

Line 178: whiteness of the concentrate decreased significantly (in which range? what is an unit for whiteness at Figure 2.c) and 3.c

Conclusion:

Can you add an information about fluorine distribution in final products after treatment of PG

Author Response

       Respond to Reviewer 2# comments

Point 1: Line 2: Please to correct "Phosphogypsume" to Phosphorgypsum.

Respond to 1: Here are our writing errors, thank you very much.

Point 2: Line 3:  Huang Jin 2 Firstname Lastname 1 (delete Firstname, LAstname)

Respond to 2: Thank you very much for your careful review. We have revised it in the article.

Point 3: Line 13: whiteness of 27.68. (what is the content of rare earth elements and radioactive elements such as thorium and uranium)

Respond to 3: Very good question. The phosphogypsum collected in this study is from Guanghan area of Deyang, Sichuan, China. According to the analysis of the phosphogypsum there, the phosphogypsum does not contain radioactive and rare earth elements. At present, phosphogypsum containing rare earth and radioactive elements is mainly distributed in Guizhou, China.

Point 4: Line 169: Agents such as DDA (please to write full name)

Respond to 4: Thank you very much for your suggestion. We have added the full name of the agents.

Point 5: Line 171: MA, and DTAC (please to write full name)

Respond to 5: Thank you very much for your suggestion. We have added the full name of the agents.

Point 6: Line 178: whiteness of the concentrate decreased significantly (in which range? what is an unit for whiteness at Figure 2.c) and 3.c

Respond to 6: Thank you again for your question. Whiteness is a dimensionless numerical value without units. In this study, a numerical value is obtained by comparative analysis between standard cards and materials to be tested by using whiteness meter.

Point 7: Conclusion: Can you add an information about fluorine distribution in final products after treatment of PG

Respond to 7: Thank you very much for your question. According to our analysis, fluorine is mainly distributed in two particle fractions of +0.15 mm and −0.025~0 mm. These two parts of materials are combined and processed separately, and mixed with other building materials.

Finally, thank you again for your careful review. Your suggestions will greatly improve the quality of our articles. We look forward to your valuable suggestions again. If you have any questions, please feel free to contact us. 

Please accept my best regards, 

Dr. Junhui Xiao 
